# Copula variational inference

**Dustin Tran**
Harvard University

**David M. Blei**
Columbia University

**Edoardo M. Airoldi**
Harvard University

## Abstract

We develop a general variational inference method that preserves dependency among the latent variables. Our method uses copulas to augment the families of distributions used in mean-field and structured approximations. Copulas model the dependency that is not captured by the original variational distribution, and thus the augmented variational family guarantees better approximations to the posterior. With stochastic optimization, inference on the augmented distribution is scalable. Furthermore, our strategy is generic: it can be applied to any inference procedure that currently uses the mean-field or structured approach. Copula variational inference has many advantages: it reduces bias; it is less sensitive to local optima; it is less sensitive to hyperparameters; and it helps characterize and interpret the dependency among the latent variables.

## 1 Introduction

Variational inference is a computationally efficient approach for approximating posterior distributions. The idea is to specify a tractable family of distributions of the latent variables and then to minimize the Kullback-Leibler divergence from it to the posterior. Combined with stochastic optimization, variational inference can scale complex statistical models to massive data sets [9, 23, 24].

Both the computational complexity and accuracy of variational inference are controlled by the factorization of the variational family. To keep optimization tractable, most algorithms use the fully-factorized family, also known as the mean-field family, where each latent variable is assumed independent. Less common, structured mean-field methods slightly relax this assumption by preserving some of the original structure among the latent variables [19]. Factorized distributions enable efficient variational inference but they sacrifice accuracy. In the exact posterior, many latent variables are dependent and mean-field methods, by construction, fail to capture this dependency.

To this end, we develop copula variational inference (COPULA VI). COPULA VI augments the traditional variational distribution with a copula, which is a flexible construction for learning dependencies in factorized distributions [3]. This strategy has many advantages over traditional VI: it reduces bias; it is less sensitive to local optima; it is less sensitive to hyperparameters; and it helps characterize and interpret the dependency among the latent variables. Variational inference has previously been restricted to either generic inference on simple models—where dependency does not make a significant difference—or writing model-specific variational updates. COPULA VI widens its applicability, providing generic inference that finds meaningful dependencies between latent variables.

In more detail, our contributions are the following.

**A generalization of the original procedure in variational inference**. COPULA VI generalizes variational inference for mean-field and structured factorizations: traditional VI corresponds to running only one step of our method. It uses coordinate descent, which monotonically decreases the KL divergence to the posterior by alternating between fitting the mean-field parameters and the copula parameters. Figure 1 illustrates COPULA VI on a toy example of fitting a bivariate Gaussian.

**Improving generic inference**. COPULA VI can be applied to any inference procedure that currently uses the mean-field or structured approach. Further, because it does not require specific knowledge

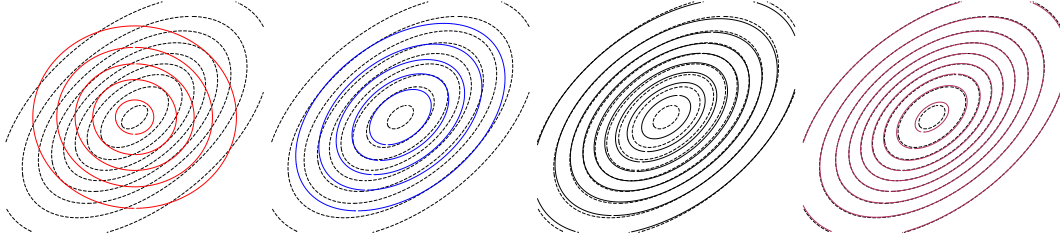

**Figure 1:** Approximations to an elliptical Gaussian. The mean-field (red) is restricted to fitting independent one-dimensional Gaussians, which is the first step in our algorithm. The second step (blue) fits a copula which models the dependency. More iterations alternate: the third refits the mean-field (green) and the fourth refits the copula (cyan), demonstrating convergence to the true posterior.

of the model, it falls into the framework of black box variational inference [15]. An investigator need only write down a function to evaluate the model log-likelihood. The rest of the algorithm's calculations, such as sampling and evaluating gradients, can be placed in a library.

**Richer variational approximations**. In experiments, we demonstrate COPULA VI on the standard example of Gaussian mixture models. We found it consistently estimates the parameters, reduces sensitivity to local optima, and reduces sensitivity to hyperparameters. We also examine how well COPULA VI captures dependencies on the latent space model [7]. COPULA VI outperforms competing methods and significantly improves upon the mean-field approximation.

## 2 Background

### 2.1 Variational inference

Let $\mathbf{x}$ be a set of observations, $\mathbf{z}$ be latent variables, and $\boldsymbol{\lambda}$ be the free parameters of a variational distribution $q(\mathbf{z}; \boldsymbol{\lambda})$. We aim to find the best approximation of the posterior $p(\mathbf{z} \mid \mathbf{x})$ using the variational distribution $q(\mathbf{z}; \boldsymbol{\lambda})$, where the quality of the approximation is measured by KL divergence. This is equivalent to maximizing the quantity

$$\mathcal{L}(\boldsymbol{\lambda}) = \mathbb{E}_{q(\mathbf{z};\boldsymbol{\lambda})}[\log p(\mathbf{x}, \mathbf{z})] - \mathbb{E}_{q(\mathbf{z};\boldsymbol{\lambda})}[\log q(\mathbf{z}; \boldsymbol{\lambda})].$$

$\mathcal{L}(\boldsymbol{\lambda})$ is the *evidence lower bound (*ELBO*)*, or the variational free energy [25]. For simpler computation, a standard choice of the variational family is a *mean-field approximation*

$$q(\mathbf{z}; \boldsymbol{\lambda}) = \prod_{i=1}^{d} q_i(\mathbf{z}_i; \boldsymbol{\lambda}_i),$$

where $\mathbf{z} = (\mathbf{z}_1, \ldots, \mathbf{z}_d)$. Note this is a strong independence assumption. More sophisticated approaches, known as *structured variational inference* [19], attempt to restore some of the dependencies among the latent variables.

In this work, we restore dependencies using copulas. Structured VI is typically tailored to individual models and is difficult to work with mathematically. Copulas learn general posterior dependencies during inference, and they do not require the investigator to know such structure in advance. Further, copulas can augment a structured factorization in order to introduce dependencies that were not considered before; thus it generalizes the procedure. We next review copulas.

### 2.2 Copulas

We will augment the mean-field distribution with a *copula*. We consider the variational family

$$q(\mathbf{z}) = \left[\prod_{i=1}^{d} q(\mathbf{z}_i)\right] c(Q(\mathbf{z}_1), \ldots, Q(\mathbf{z}_d)).$$

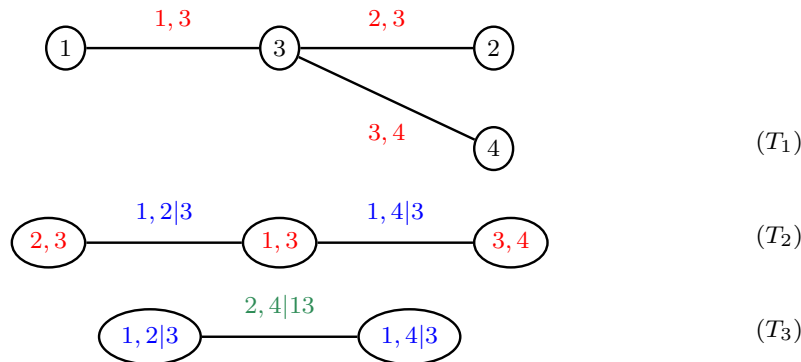

**Figure 2:** Example of a vine $\mathcal{V}$ which factorizes a copula density of four random variables $c(\mathbf{u}_1, \mathbf{u}_2, \mathbf{u}_3, \mathbf{u}_4)$ into a product of 6 pair copulas. Edges in the tree $T_j$ are the nodes of the lower level tree $T_{j+1}$, and each edge determines a bivariate copula which is conditioned on all random variables that its two connected nodes share.

Here $Q(\mathbf{z}_i)$ is the marginal cumulative distribution function (CDF) of the random variable $\mathbf{z}_i$, and $c$ is a joint distribution of $[0, 1]$ random variables.[1] The distribution $c$ is called a copula of $\mathbf{z}$: it is a joint multivariate density of $Q(\mathbf{z}_1), \ldots, Q(\mathbf{z}_d)$ with uniform marginal distributions [21]. For any distribution, a factorization into a product of marginal densities and a copula always exists and integrates to one [14].

Intuitively, the copula captures the information about the multivariate random variable after eliminating the marginal information, i.e., by applying the probability integral transform on each variable. The copula captures only and all of the dependencies among the $\mathbf{z}_i$'s. Recall that, for all random variables, $Q(\mathbf{z}_i)$ is uniform distributed. Thus the marginals of the copula give no information.

For example, the bivariate Gaussian copula is defined as

$$c(\mathbf{u}_1, \mathbf{u}_2; \rho) = \Phi_\rho(\Phi^{-1}(\mathbf{u}_1), \Phi^{-1}(\mathbf{u}_2)).$$

If $\mathbf{u}_1, \mathbf{u}_2$ are independent uniform distributed, the inverse CDF $\Phi^{-1}$ of the standard normal transforms $(\mathbf{u}_1, \mathbf{u}_2)$ to independent normals. The CDF $\Phi_\rho$ of the bivariate Gaussian distribution, with mean zero and Pearson correlation $\rho$, squashes the transformed values back to the unit square. Thus the Gaussian copula directly correlates $\mathbf{u}_1$ and $\mathbf{u}_2$ with the Pearson correlation parameter $\rho$.

### 2.2.1 Vine copulas

It is difficult to specify a copula. We must find a family of distributions that is easy to compute with and able to express a broad range of dependencies. Much work focuses on two-dimensional copulas, such as the Student-$t$, Clayton, Gumbel, Frank, and Joe copulas [14]. However, their multivariate extensions do not flexibly model dependencies in higher dimensions [4]. Rather, a successful approach in recent literature has been by combining sets of conditional bivariate copulas; the resulting joint is called a *vine* [10, 13].

A vine $\mathcal{V}$ factorizes a copula density $c(\mathbf{u}_1, \ldots, \mathbf{u}_d)$ into a product of conditional bivariate copulas, also called pair copulas. This makes it easy to specify a high-dimensional copula. One need only express the dependence for each pair of random variables conditioned on a subset of the others.

Figure 2 is an example of a vine which factorizes a 4-dimensional copula into the product of 6 pair copulas. The first tree $T_1$ has nodes $1, 2, 3, 4$ representing the random variables $\mathbf{u}_1, \mathbf{u}_2, \mathbf{u}_3, \mathbf{u}_4$ respectively. An edge corresponds to a pair copula, e.g., $1, 4$ symbolizes $c(\mathbf{u}_1, \mathbf{u}_4)$. Edges in $T_1$ collapse into nodes in the next tree $T_2$, and edges in $T_2$ correspond to conditional bivariate copulas, e.g., $1, 2|3$ symbolizes $c(\mathbf{u}_1, \mathbf{u}_2|\mathbf{u}_3)$. This proceeds to the last nested tree $T_3$, where $2, 4|13$ symbolizes

$c(\mathbf{u}_2, \mathbf{u}_4 | \mathbf{u}_1, \mathbf{u}_3)$. The vine structure specifies a complete factorization of the multivariate copula, and each pair copula can be of a different family with its own set of parameters:

$$c(\mathbf{u}_1, \mathbf{u}_2, \mathbf{u}_3, \mathbf{u}_4) = \Big[ c(\mathbf{u}_1, \mathbf{u}_3) c(\mathbf{u}_2, \mathbf{u}_3) c(\mathbf{u}_3, \mathbf{u}_4) \Big] \Big[ c(\mathbf{u}_1, \mathbf{u}_2 | \mathbf{u}_3) c(\mathbf{u}_1, \mathbf{u}_4 | \mathbf{u}_3) \Big] \Big[ c(\mathbf{u}_2, \mathbf{u}_4 | \mathbf{u}_1, \mathbf{u}_3) \Big].$$

Formally, a vine is a nested set of trees $\mathcal{V} = \{T_1, \ldots, T_{d-1}\}$ with the following properties:

1. Tree $T_j = \{N_j, E_j\}$ has $d + 1 - j$ nodes and $d - j$ edges.

2. Edges in the $j^{th}$ tree $E_j$ are the nodes in the $(j + 1)^{th}$ tree $N_{j+1}$.

3. Two nodes in tree $T_{j+1}$ are joined by an edge only if the corresponding edges in tree $T_j$ share a node.

Each edge $e$ in the nested set of trees $\{T_1, \ldots, T_{d-1}\}$ specifies a different pair copula, and the product of all edges comprise of a factorization of the copula density. Since there are a total of $d(d-1)/2$ edges, $\mathcal{V}$ factorizes $c(\mathbf{u}_1, \ldots, \mathbf{u}_d)$ as the product of $d(d-1)/2$ pair copulas.

Each edge $e(i, k) \in T_j$ has a *conditioning set* $D(e)$, which is a set of variable indices $1, \ldots, d$. We define $c_{ik|D(e)}$ to be the bivariate copula density for $\mathbf{u}_i$ and $\mathbf{u}_k$ given its conditioning set:

$$c_{ik|D(e)} = c\Big( Q(\mathbf{u}_i | \mathbf{u}_j : j \in D(e)), Q(\mathbf{u}_i | \mathbf{u}_j : j \in D(e)) \Big| \mathbf{u}_j : j \in D(e) \Big). \tag{1}$$

Both the copula and the CDF's in its arguments are conditional on $D(e)$. A vine specifies a factorization of the copula, which is a product over all edges in the $d - 1$ levels:

$$c(\mathbf{u}_1, \ldots, \mathbf{u}_d; \boldsymbol{\eta}) = \prod_{j=1}^{d-1} \prod_{e(i,k) \in E_j} c_{ik|D(e)}. \tag{2}$$

We highlight that $c$ depends on $\boldsymbol{\eta}$, the set of all parameters to the pair copulas. The vine construction provides us with the flexibility to model dependencies in high dimensions using a decomposition of pair copulas which are easier to estimate. As we shall see, the construction also leads to efficient stochastic gradients by taking individual (and thus easy) gradients on each pair copula.

## 3 Copula variational inference

We now introduce *copula variational inference (*COPULA VI*)*, our method for performing accurate and scalable variational inference. For simplicity, consider the mean-field factorization augmented with a copula (we later extend to structured factorizations). The copula-augmented variational family is

$$q(\mathbf{z}; \boldsymbol{\lambda}, \boldsymbol{\eta}) = \underbrace{\left[ \prod_{i=1}^d q(\mathbf{z}_i; \boldsymbol{\lambda}) \right]}_{\text{mean-field}} \underbrace{c(Q(\mathbf{z}_1; \boldsymbol{\lambda}), \ldots, Q(\mathbf{z}_d; \boldsymbol{\lambda}); \boldsymbol{\eta})}_{\text{copula}}, \tag{3}$$

where $\boldsymbol{\lambda}$ denotes the mean-field parameters and $\boldsymbol{\eta}$ the copula parameters. With this family, we maximize the augmented ELBO,

$$\mathcal{L}(\boldsymbol{\lambda}, \boldsymbol{\eta}) = \mathbb{E}_{q(\mathbf{z}; \boldsymbol{\lambda}, \boldsymbol{\eta})}[\log p(\mathbf{x}, \mathbf{z})] - \mathbb{E}_{q(\mathbf{z}; \boldsymbol{\lambda}, \boldsymbol{\eta})}[\log q(\mathbf{z}; \boldsymbol{\lambda}, \boldsymbol{\eta})].$$

COPULA VI alternates between two steps: 1) fix the copula parameters $\boldsymbol{\eta}$ and solve for the mean-field parameters $\boldsymbol{\lambda}$; and 2) fix the mean-field parameters $\boldsymbol{\lambda}$ and solve for the copula parameters $\boldsymbol{\eta}$. This generalizes the mean-field approximation, which is the special case of initializing the copula to be uniform and stopping after the first step. We apply stochastic approximations [18] for each step with gradients derived in the next section. We set the learning rate $\rho_t \in \mathbb{R}$ to satisfy a Robbins-Monro schedule, i.e., $\sum_{t=1}^{\infty} \rho_t = \infty$, $\sum_{t=1}^{\infty} \rho_t^2 < \infty$. A summary is outlined in Algorithm 1.

This alternating set of optimizations falls in the class of minorize-maximization methods, which includes many procedures such as the EM algorithm [1], the alternating least squares algorithm, and the iterative procedure for the generalized method of moments. Each step of COPULA VI monotonically increases the objective function and therefore better approximates the posterior distribution.

---

**Algorithm 1:** Copula variational inference (COPULA VI)

---

**Input**: Data $\mathbf{x}$, Model $p(\mathbf{x}, \mathbf{z})$, Variational family $q$.

Initialize $\boldsymbol{\lambda}$ randomly, $\boldsymbol{\eta}$ so that $c$ is uniform.

**while** *change in ELBO is above some threshold* **do**

    // Fix $\boldsymbol{\eta}$, maximize over $\boldsymbol{\lambda}$.

    Set iteration counter $t = 1$.

    **while** *not converged* **do**

        Draw sample $\mathbf{u} \sim \mathrm{Unif}([0, 1]^d)$.

        Update $\boldsymbol{\lambda} = \boldsymbol{\lambda} + \rho_t \nabla_{\boldsymbol{\lambda}} \mathcal{L}$. (Eq.5, Eq.6)

        Increment $t$.

    **end**

    // Fix $\boldsymbol{\lambda}$, maximize over $\boldsymbol{\eta}$.

    Set iteration counter $t = 1$.

    **while** *not converged* **do**

        Draw sample $\mathbf{u} \sim \mathrm{Unif}([0, 1]^d)$.

        Update $\boldsymbol{\eta} = \boldsymbol{\eta} + \rho_t \nabla_{\boldsymbol{\eta}} \mathcal{L}$. (Eq.7)

        Increment $t$.

    **end**

**end**

**Output**: Variational parameters $(\boldsymbol{\lambda}, \boldsymbol{\eta})$.

---

COPULA VI has the same generic input requirements as black-box variational inference [15]—the user need only specify the joint model $p(\mathbf{x}, \mathbf{z})$ in order to perform inference. Further, copula variational inference easily extends to the case when the original variational family uses a structured factorization. By the vine construction, one simply fixes pair copulas corresponding to pre-existent dependence in the factorization to be the independence copula. This enables the copula to only model dependence where it does not already exist.

Throughout the optimization, we will assume that the tree structure and copula families are given and fixed. We note, however, that these can be learned. In our study, we learn the tree structure using sequential tree selection [2] and learn the families, among a choice of 16 bivariate families, through Bayesian model selection [6] (see supplement). In preliminary studies, we've found that re-selection of the tree structure and copula families do not significantly change in future iterations.

### 3.1 Stochastic gradients of the ELBO

To perform stochastic optimization, we require stochastic gradients of the ELBO with respect to both the mean-field and copula parameters. The COPULA VI objective leads to efficient stochastic gradients and with low variance.

We first derive the gradient with respect to the mean-field parameters. In general, we can apply the score function estimator [15], which leads to the gradient

$$\nabla_{\boldsymbol{\lambda}} \mathcal{L} = \mathbb{E}_{q(\mathbf{z};\boldsymbol{\lambda},\boldsymbol{\eta})}[\nabla_{\boldsymbol{\lambda}} \log q(\mathbf{z};\boldsymbol{\lambda},\boldsymbol{\eta}) \cdot (\log p(\mathbf{x}, \mathbf{z}) - \log q(\mathbf{z};\boldsymbol{\lambda},\boldsymbol{\eta}))]. \tag{4}$$

We follow noisy unbiased estimates of this gradient by sampling from $q(\cdot)$ and evaluating the inner expression. We apply this gradient for discrete latent variables.

When the latent variables $\mathbf{z}$ are differentiable, we use the reparameterization trick [17] to take advantage of first-order information from the model, i.e., $\nabla_{\mathbf{z}} \log p(\mathbf{x}, \mathbf{z})$. Specifically, we rewrite the expectation in terms of a random variable $\mathbf{u}$ such that its distribution $s(\mathbf{u})$ does not depend on the variational parameters and such that the latent variables are a deterministic function of $\mathbf{u}$ and the mean-field parameters, $\mathbf{z} = \mathbf{z}(\mathbf{u}; \boldsymbol{\lambda})$. Following this reparameterization, the gradients propagate

inside the expectation,
$$\nabla_{\boldsymbol{\lambda}}\mathcal{L} = \mathbb{E}_{s(\mathbf{u})}[(\nabla_{\mathbf{z}}\log p(\mathbf{x},\mathbf{z}) - \nabla_{\mathbf{z}}\log q(\mathbf{z};\boldsymbol{\lambda},\boldsymbol{\eta}))\nabla_{\boldsymbol{\lambda}}\mathbf{z}(\mathbf{u};\boldsymbol{\lambda})]. \tag{5}$$
This estimator reduces the variance of the stochastic gradients [17]. Furthermore, with a copula variational family, this type of reparameterization using a uniform random variable $\mathbf{u}$ and a deterministic function $\mathbf{z} = \mathbf{z}(\mathbf{u};\boldsymbol{\lambda},\boldsymbol{\eta})$ is always possible. (See the supplement.)

The reparameterized gradient (Eq.5) requires calculation of the terms $\nabla_{\mathbf{z}_i}\log q(\mathbf{z};\boldsymbol{\lambda},\boldsymbol{\eta})$ and $\nabla_{\boldsymbol{\lambda}_i}\mathbf{z}(\mathbf{u};\boldsymbol{\lambda},\boldsymbol{\eta})$ for each $i$. The latter is tractable and derived in the supplement; the former decomposes as

$$\nabla_{\mathbf{z}_i}\log q(\mathbf{z};\boldsymbol{\lambda},\boldsymbol{\eta}) = \nabla_{\mathbf{z}_i}\log q(\mathbf{z}_i;\boldsymbol{\lambda}_i) + \nabla_{Q(\mathbf{z}_i;\boldsymbol{\lambda}_i)}\log c(Q(\mathbf{z}_1;\boldsymbol{\lambda}_1),\dots,Q(\mathbf{z}_d;\boldsymbol{\lambda}_d);\boldsymbol{\eta})\nabla_{\mathbf{z}_i}Q(\mathbf{z}_i;\boldsymbol{\lambda}_i)$$

$$= \nabla_{\mathbf{z}_i}\log q(\mathbf{z}_i;\boldsymbol{\lambda}_i) + q(\mathbf{z}_i;\boldsymbol{\lambda}_i)\sum_{j=1}^{d-1}\sum_{\substack{e(k,\ell)\in E_j:\\ i\in\{k,\ell\}}}\nabla_{Q(\mathbf{z}_i;\boldsymbol{\lambda}_i)}\log c_{k\ell|D(e)}. \tag{6}$$

The summation in Eq.6 is over all pair copulas which contain $Q(\mathbf{z}_i;\boldsymbol{\lambda}_i)$ as an argument. In other words, the gradient of a latent variable $\mathbf{z}_i$ is evaluated over both the marginal $q(\mathbf{z}_i)$ and all pair copulas which model correlation between $\mathbf{z}_i$ and any other latent variable $\mathbf{z}_j$. A similar derivation holds for calculating terms in the score function estimator.

We now turn to the gradient with respect to the copula parameters. We consider copulas which are differentiable with respect to their parameters. This enables an efficient reparameterized gradient

$$\nabla_{\boldsymbol{\eta}}\mathcal{L} = \mathbb{E}_{s(\mathbf{u})}[(\nabla_{\mathbf{z}}\log p(\mathbf{x},\mathbf{z}) - \nabla_{\mathbf{z}}\log q(\mathbf{z};\boldsymbol{\lambda},\boldsymbol{\eta}))\nabla_{\boldsymbol{\eta}}\mathbf{z}(\mathbf{u};\boldsymbol{\lambda},\boldsymbol{\eta})]. \tag{7}$$
The requirements are the same as for the mean-field parameters.

Finally, we note that the only requirement on the model is the gradient $\nabla_{\mathbf{z}}\log p(\mathbf{x},\mathbf{z})$. This can be calculated using automatic differentiation tools [22]. Thus COPULA VI can be implemented in a library and applied without requiring any manual derivations from the user.

## 3.2 Computational complexity

In the vine factorization of the copula, there are $d(d-1)/2$ pair copulas, where $d$ is the number of latent variables. Thus stochastic gradients of the mean-field parameters $\boldsymbol{\lambda}$ and copula parameters $\boldsymbol{\eta}$ require $\mathcal{O}(d^2)$ complexity. More generally, one can apply a low rank approximation to the copula by truncating the number of levels in the vine (see Figure 2). This reduces the number of pair copulas to be $Kd$ for some $K > 0$, and leads to a computational complexity of $\mathcal{O}(Kd)$.

Using sequential tree selection for learning the vine structure [2], the most correlated variables are at the highest level of the vines. Thus a truncated low rank copula only forgets the weakest correlations. This generalizes low rank Gaussian approximations, which also have $\mathcal{O}(Kd)$ complexity [20]: it is the special case when the mean-field distribution is the product of independent Gaussians, and each pair copula is a Gaussian copula.

## 3.3 Related work

Preserving structure in variational inference was first studied by Saul and Jordan [19] in the case of probabilistic neural networks. It has been revisited recently for the case of conditionally conjugate exponential families [8]. Our work differs from this line in that we learn the dependency structure during inference, and thus we do not require explicit knowledge of the model. Further, our augmentation strategy works more broadly to any posterior distribution and any factorized variational family, and thus it generalizes these approaches.

A similar augmentation strategy is higher-order mean-field methods, which are a Taylor series correction based on the difference between the posterior and its mean-field approximation [11]. Recently, Giordano et al. [5] consider a covariance correction from the mean-field estimates. All these methods assume the mean-field approximation is reliable for the Taylor series expansion to make sense, which is not true in general and thus is not robust in a black box framework. Our approach alternates the estimation of the mean-field and copula, which we find empirically leads to more robust estimates than estimating them simultaneously, and which is less sensitive to the quality of the mean-field approximation.

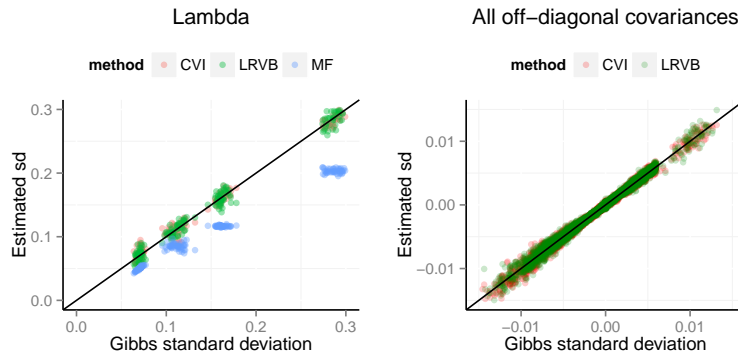

**Figure 3:** Covariance estimates from copula variational inference (COPULA VI), mean-field (MF), and linear response variational Bayes (LRVB) to the ground truth (Gibbs samples). COPULA VI and LRVB effectively capture dependence while MF underestimates variance and forgets covariances.

## 4 Experiments

We study COPULA VI with two models: Gaussian mixtures and the latent space model [7]. The Gaussian mixture is a classical example of a model for which it is difficult to capture posterior dependencies. The latent space model is a modern Bayesian model for which the mean-field approximation gives poor estimates of the posterior, and where modeling posterior dependencies is crucial for uncovering patterns in the data.

There are several implementation details of COPULA VI. At each iteration, we form a stochastic gradient by generating $m$ samples from the variational distribution and taking the average gradient. We set $m = 1024$ and follow asynchronous updates [16]. We set the step-size using ADAM [12].

### 4.1 Mixture of Gaussians

We follow the goal of Giordano et al. [5], which is to estimate the posterior covariance for a Gaussian mixture. The hidden variables are a $K$-vector of mixture proportions $\boldsymbol{\pi}$ and a set of $K$ $P$-dimensional multivariate normals $\mathcal{N}(\boldsymbol{\mu}_k, \boldsymbol{\Lambda}_k^{-1})$, each with unknown mean $\boldsymbol{\mu}_k$ (a $P$-vector) and $P \times P$ precision matrix $\boldsymbol{\Lambda}_k$. In a mixture of Gaussians, the joint probability is

$$p(\mathbf{x}, \mathbf{z}, \boldsymbol{\mu}, \boldsymbol{\Lambda}^{-1}, \boldsymbol{\pi}) = p(\boldsymbol{\pi}) \prod_{k=1}^{K} p(\boldsymbol{\mu}_k, \boldsymbol{\Lambda}_k^{-1}) \prod_{n=1}^{N} p(\mathbf{x}_n \mid \mathbf{z}_n, \boldsymbol{\mu}_{\mathbf{z}_n}, \boldsymbol{\Lambda}_{\mathbf{z}_n}^{-1}) p(\mathbf{z}_n \mid \boldsymbol{\pi}),$$

with a Dirichlet prior $p(\boldsymbol{\pi})$ and a normal-Wishart prior $p(\boldsymbol{\mu}_k, \boldsymbol{\Lambda}_k^{-1})$.

We first apply the mean-field approximation (MF), which assigns independent factors to $\boldsymbol{\mu}, \boldsymbol{\pi}, \boldsymbol{\Lambda}$, and $\mathbf{z}$. We then perform COPULA VI over the copula-augmented mean-field distribution, i.e., one which includes pair copulas over the latent variables. We also compare our results to linear response variational Bayes (LRVB) [5], which is a posthoc correction technique for covariance estimation in variational inference. Higher-order mean-field methods demonstrate similar behavior as LRVB. Comparisons to structured approximations are omitted as they require explicit factorizations and are not black box. Standard black box variational inference [15] corresponds to the MF approximation.

We simulate 10,000 samples with $K = 2$ components and $P = 2$ dimensional Gaussians. Figure 3 displays estimates for the standard deviations of $\boldsymbol{\Lambda}$ for 100 simulations, and plots them against the ground truth using 500 effective Gibb samples. The second plot displays all off-diagonal covariance estimates. Estimates for $\boldsymbol{\mu}$ and $\boldsymbol{\pi}$ indicate the same pattern and are given in the supplement.

When initializing at the true mean-field parameters, both COPULA VI and LRVB achieve consistent estimates of the posterior variance. MF underestimates the variance, which is a well-known limitation [25]. Note that because the MF estimates are initialized at the truth, COPULA VI converges to the true posterior upon one step of fitting the copula. It does not require alternating more steps.

| Variational inference methods | Predictive Likelihood | Runtime |
|---|---|---|
| Mean-field | -383.2 | 15 min. |
| LRVB | -330.5 | 38 min. |
| COPULA VI (2 steps) | -303.2 | 32 min. |
| COPULA VI (5 steps) | -80.2 | 1 hr. 17 min. |
| COPULA VI (converged) | -50.5 | 2 hr. |

**Table 1:** Predictive likelihood on the latent space model. Each COPULA VI step either refits the mean-field or the copula. COPULA VI converges in roughly 10 steps and already significantly outperforms both mean-field and LRVB upon fitting the copula once (2 steps).

COPULA VI is more robust than LRVB. As a toy demonstration, we analyze the MNIST data set of handwritten digits, using 12,665 training examples and 2,115 test examples of 0's and 1's. We perform "unsupervised" classification, i.e., classify without using training labels: we apply a mixture of Gaussians to cluster, and then classify a digit based on its membership assignment. COPULA VI reports a test set error rate of 0.06, whereas LRVB ranges between 0.06 and 0.32 depending on the mean-field estimates. LRVB and similar higher order mean-field methods correct an existing MF solution—it is thus sensitive to local optima and the general quality of that solution. On the other hand, COPULA VI re-adjusts both the MF and copula parameters as it fits, making it more robust to initialization.

### 4.2 Latent space model

We next study inference on the latent space model [7], a Bernoulli latent factor model for network analysis. Each node in an $N$-node network is associated with a $P$-dimensional latent variable $\mathbf{z} \sim N(\boldsymbol{\mu}, \boldsymbol{\Lambda}^{-1})$. Edges between pairs of nodes are observed with high probability if the nodes are close to each other in the latent space. Formally, an edge for each pair $(i, j)$ is observed with probability $\text{logit}(p) = \theta - |\mathbf{z}_i - \mathbf{z}_j|$, where $\theta$ is a model parameter.

We generate an $N = 100,000$ node network with latent node attributes from a $P = 10$ dimensional Gaussian. We learn the posterior of the latent attributes in order to predict the likelihood of held-out edges. MF applies independent factors on $\boldsymbol{\mu}, \boldsymbol{\Lambda}, \theta$ and $\mathbf{z}$, LRVB applies a correction, and COPULA VI uses the fully dependent variational distribution. Table 1 displays the likelihood of held-out edges and runtime. We also attempted Hamiltonian Monte Carlo but it did not converge after five hours.

COPULA VI dominates other methods in accuracy upon convergence, and the copula estimation without refitting (2 steps) already dominates LRVB in both runtime and accuracy. We note however that LRVB requires one to invert a $\mathcal{O}(NK^3) \times \mathcal{O}(NK^3)$ matrix. We can better scale the method and achieve faster estimates than COPULA VI if we applied stochastic approximations for the inversion. However, COPULA VI always outperforms LRVB and is still fast on this 100,000 node network.

## 5 Conclusion

We developed copula variational inference (COPULA VI). COPULA VI is a new variational inference algorithm that augments the mean-field variational distribution with a copula; it captures posterior dependencies among the latent variables. We derived a scalable and generic algorithm for performing inference with this expressive variational distribution. We found that COPULA VI significantly reduces the bias of the mean-field approximation, better estimates the posterior variance, and is more accurate than other forms of capturing posterior dependency in variational approximations.

**Acknowledgments**

We thank Luke Bornn, Robin Gong, and Alp Kucukelbir for their insightful comments. This work is supported by NSF IIS-0745520, IIS-1247664, IIS-1009542, ONR N00014-11-1-0651, DARPA FA8750-14-2-0009, N66001-15-C-4032, Facebook, Adobe, Amazon, and the John Templeton Foundation.

## Footnotes

[1]We overload the notation for the marginal CDF $Q$ to depend on the names of the argument, though we occasionally use $Q_i(\mathbf{z}_i)$ when more clarity is needed. This is analogous to the standard convention of overloading the probability density function $q(\cdot)$.

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
