[Supplementary Material · nips2015_supplement.pdf]

# Supplement: Copula variational inference

**Dustin Tran**
Harvard University

**David M. Blei**
Columbia University

**Edoardo M. Airoldi**
Harvard University

## 1 Sampling from the copula-augmented variational distribution

We sample from the copula-augmented distribution by repeatedly doing inverse transform sampling [1], also known as inverse CDF, on the individual pair copulas and finally the marginals. More specifically, the sampling procedure is as follows:

1. Generate $\mathbf{u} = (\mathbf{u}_1, \dots, \mathbf{u}_d)$ where each $\mathbf{u}_i \sim \mathcal{U}(0, 1)$.

2. Calculate $\mathbf{v} = (\mathbf{v}_1, \dots, \mathbf{v}_d)$ which follows a joint uniform distribution with dependencies given by the copula:

$$\mathbf{v}_1 = \mathbf{u}_1$$
$$\mathbf{v}_2 = Q_{2\,|\,1}^{-1}(\mathbf{u}_2\,|\,\mathbf{v}_1)$$
$$\mathbf{v}_3 = Q_{3\,|\,12}^{-1}(\mathbf{u}_3\,|\,\mathbf{v}_1, \mathbf{v}_2)$$
$$\vdots$$
$$\mathbf{v}_d = Q_{d\,|\,12\cdots d-1}^{-1}(\mathbf{u}_d\,|\,\mathbf{v}_1, \mathbf{v}_2, \dots, \mathbf{v}_{d-1})$$

   Explicit calculations of the inverse of the conditional CDFs $Q_{i|12\cdots i-1}^{-1}$ can be found in Kurowicka and Cooke [3]. The procedure loops through the $d(d-1)/2$ pair copulas and thus has worst-case complexity of $\mathcal{O}(d^2)$.

3. Calculate $\mathbf{z} = (Q_1^{-1}(\mathbf{v}_1), \dots, Q_d^{-1}(\mathbf{v}_d))$, which is a sample from the copula-augmented distribution $q(\mathbf{z}; \boldsymbol{\lambda}, \boldsymbol{\eta})$.

Evaluating gradients with respect to $\boldsymbol{\lambda}$ and $\boldsymbol{\eta}$ easily follows from backpropagation, i.e., by applying the chain rule on this sequence of deterministic transformations.

## 2 Choosing the tree structure and pair copula families

We assume that the vine structure and pair copula families are specified in order to perform copula variational inference (COPULA VI), in the same way one must specify the mean-field family for black box variational inference [5]. In general however, given a factorization of the variational distribution, one can determine the tree structure and pair copula families based on synthetic data of the latent variables $z$.

During tree selection, enumerating and calculating all possibilities is computationally intractable, as the number of possible vines on $d$ variables grows factorially: there exist $d!/2 \cdot 2^{\binom{d-2}{2}}$ many choices [4]. The most common approach in practice is to sequentially select the maximum spanning tree starting from the initial tree $T_1$, where the weights of an edge are assigned by absolute values of the Kendall's $\tau$ correlation on each pair of random variables. Intuitively, the tree structures are selected as to model the strongest pairwise dependencies. This procedure of sequential tree selection follows Dissmann et al. [2].

In order to select a family of distributions for each conditional bivariate copula in the vine, one may employ Bayesian model selection, i.e., choose among a set of families which maximizes the marginal

likelihood. We note that both the sequential tree selection and model selection are implemented in the `VineCopula` package in R [6], which makes it easy for users to learn the structure and families for the copula-augmented variational distribution.

We also list below the 16 bivariate copula families used in our experiments.

| Family | Parameter | $\theta(\tau)$ |
|---|---|---|
| **Independent** | — | — |
| Gaussian | $\theta \in [-1, 1]$ | $\sin\left(\dfrac{\pi}{2}\tau\right)$ |
| Student-$t$ | $\theta \in [-1, 1]$ | |
| Clayton | $\theta \in (0, \infty)$ | $2\tau/(1 - \tau)$ |
| Gumbel | $\theta \in [1, \infty)$ | $1/(1 - \tau)$ |
| Frank | $\theta \in (0, \infty)$ | No closed form |
| Joe | $\theta \in (1, \infty)$ | |

**Table 1:** The 16 bivariate copula families, with their parameter domains and expressed in terms of Kendall's $\tau$ correlations, that we consider in experiments. We include rotated versions ($90°$, $180°$, and $270°$) of the Clayton, Gumbel, and Joe copulas.

**Figure 1:** Example of a Frank copula with correlation parameter 0.8, which is used to model weak symmetric tail dependencies.

## 3   Additional Gaussian mixture experiments

We include figures showing the standard deviation estimates for $\boldsymbol{\mu}$ and $\boldsymbol{\pi}$ which were not included in the main paper. The results indicate the same pattern as for $\boldsymbol{\Lambda}$.

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

**Figure 2:** Covariance estimates from copula variational inference (COPULA VI), mean-field (mean-field (MF)), and linear response variational Bayes (linear response variational Bayes (LRVB)) to the ground truth (Gibbs samples). COPULA VI and LRVB effectively capture dependence while MF under-estimates variance and forgets covariances.