[Reviews · NeurIPS 2015]

Submitted by Assigned_Reviewer_1

The approach being taken here is to use copulas to capture dependencies between units which are missed by mean field. This has some strong similarities to higher order mean field techniques, where higher and higher order dependencies between units are captured by additional terms in a mean field expansion. Should include some discussion of this relationship / cite it as another approach for capturing higher order dependencies.

Abstract - make explicit that the problem you are addressing is inference over latent variables

37 - 'corresponding to running only one step of the algorithm' this doesn't make sense yet here. Expand on it here, or save it until you've given context.

44 - I have a strong skepticism of this as a black box method. Copula methods (I believe even the vine technique you use) typically remove dependencies along one axis in state space at a time. This is great if the problem is low dimensional or has a known structure with low order or simple hierarchical interactions. In general though, there will be an exponential number of dependencies between units, and each time state space is rescaled using the CDF along one axis in order to remove one dependency, the rest of the dependencies are distorted and made more complex.

110 - Flip the top row in figure left-right so that the graph is oriented the same way in each row.

142-145 - Confusing. rho is a ?measured? correlation capturing pariwise dependencies? And in the case of a 2d Gaussian will lead to an exact model match?

Section 2.2.1 - How is the edge structure chosen for the graph used for the vine? I would imagine this is crucial to performance and application as a black box method.

183 - conditioning on D(e). Doesn't the size of the state space you are conditioning on increase exponentially with depth? Is there a function for the dependence? Or a shallow maximum depth?

I think layer T_j is acting on the outputs of layer T_{j+1}, i.e. after the units have already been transformed by T_j? I found this confusing in the text though.

185 - 'rigorizes' -> 'makes rigorous'

238 - think this should be 'minorization-maximization'

251 - 'the restriction' this is not a restriction, just a common choice

270 - where did v come from? Is it just a new variable name being used to hold sample output?

Section 3.2 - This section could be simplified, to take home message and gradient formula, with details moved to appendix.

Figure 3 - Use smaller points (and maybe alpha) so can see both distributions when they overlap. Or mix plotting order so the colors don't occur in occluding layers.

Table 1 - Wouldn't comparisons to black box variational inference methods be more appropriate here? Or higher order mean field methods like TAP?

To me all the examples in the experimental section seem very low dimensional. I'm very curious how performance scales with the dimensionality of the state space. Can you say anything about this?

419 - That HMC could not be made to sample from this 10-dimensional latent distribution over many hours seems *very* difficult to believe. How did you identify non-convergence? How did you choose the hyperparameters? What implementation did you use?
Summary: The paper presents a method to iteratively use copulas to improve mean field (or other structured) approximations to a distribution. The underlying idea seems sensible, and the writing style was good. I was confused or skeptical about the details, and the experiments were not very convincing.

Submitted by Assigned_Reviewer_2

I like the basic idea of the paper;

using copulas to restore some of the dependencies which are broken in mean field variational Bayes seems like a good idea and the use of vine copulas seems to provide some useful efficiencies in the computations.

However, whether this generalizes well to high dimensions depends very much on whether the large number of parameters describing the dependence structure can be efficiently learnt.

Perhaps some of these details are explained in the supplementary material, but the supplementary material available on CMT is for a completely different paper so I'm unable to say.

The numerical experiments are a bit limited as they stand.

There is a low-dimensional example involving mixtures and a very brief description of a more interesting example involving a latent space model.

In both cases non-independence copulas are used to describe dependence involving covariance matrix parameters and I wondered whether the positive definiteness constraint was difficult to handle within the copula framework - is some kind of reparametrization required for that?

If I've understood correctly in the latent space model a copula is learnt to describe dependencies within each block of latent node attributes as well as for the mean and covariance matrix of node attributes so the number of copula parameters learnt here is large.

I'm not sure exactly how the comparisons with LRVB are done;

LRVB after all simply gives an improved estimate of the posterior covariance matrix.

For non-Gaussian variational factors I wasn't sure precisely how that was used to derive an improved predictive inference that could reasonably be compared with the CVI approach.

So I'm not clear on what was done here for the classification in the MNIST data for the two component mixture example and how the LRVB was used to get a comparable predictive likelihood to CVI for the latent space model.

Summary: This paper proposes a way of learning an appropriate dependence structure for variational approximations using copulas.

The methodology seems highly original and quite general but I did find details of some of the numerical experiments a bit unclear.

Submitted by Assigned_Reviewer_3

The typical mean-field approximation made in variational inference results in a well-known range of issues, from poor approximation of the distribution as a whole to poor covariance estimates in particular. This is a pressing problem for variational inference, and the authors aim to address it by introducing copulas into the variational approximating distribution. The authors' experimental results suggest that this may be a promising direction for improving on variational approximations. However, some details of the method are not covered as much as desired, and the experiments seem somewhat limited.

Major points

(1) In order to run their method, the authors need to choose a vine and a copula. While the authors spend a little time discussing that they choose these according to an algorithm, it's not clear what the approximative significance of the vine in particular is. Why are some vines better than others? Why are some copulas better than others? Are all possible vines considered? Is that problematic in many-parameter models? How are the possible copulas chosen? Is there any reason to think these choices of copulas are enough?

E.g., on the tree point,

p. 4: line 163--168: Where does the first tree come from? Can it be any tree on the model variables? Does it matter?

p. 5, line 243: What does it mean to learn the tree structure? What is there to learn?

(2) The authors seem to use one model per performance metric: a Gaussian mixture model for variance modeling and a latent state space model for predictive likelihood. (Why not compare both models on both metrics?) This isn't necessarily bad; it could be a starting point for future exploration. But then the authors seem to make somewhat broad conclusions based on one model each. It's also unclear why the authors plot just one particular set of parameters from the Gaussian mixture model in Figure 3. Aren't there are other mixture model parameters (like the component means in Eq. 18)? For that matter, what are the hyperparameters in the each generative model (the ones used in the experiment)?

p. 8, line 393--396: It seems from Fig 3 that CVI and LRVB both estimate variances equally well on this problem. LRVB focuses on covariances. Presumably the point of CVI is to get an entire posterior that is better. Table 1 gets at this a little, but it would be nice to see more comparisons on, say, estimates of other posterior statistics than just covariances. Or on other models. And why not compare to structured mean-field or other recent methods that get beyond mean-field for predictive likelihood?

p. 8, line 418: "CVI always outperforms LRVB". This statement seems a bit broad to be warranted from the experiments presented. The authors might reasonably say CVI outperforms LRVB in predictive likelihood (based on Table 1)---but does it outperform across other models besides just the one mixture model presented? Or do the authors really mean to say it outperforms LRVB in other ways? It is unclear how the two compare in covariance estimation (besides looking roughly the same in Figure 3).

Minor points

p. 1, line 35--37: This sentence is a little difficult to parse on the first pass. It sounds like the authors are saying it's a generalization but also a special case. Presumably they are just saying that their algorithm iterates between mean-field steps and copula steps. Does "structured factorizations" here refer to [19]? If so, it would help the reader to be explicit.

p. 2, line 83: It seems a little unusual of a notational choice to use z for all latent variables; it seems to be usually reserved for indicators

p. 2, line 84: KL is technically not a distance since it is not symmetric

p. 3, line 143: "the bivariate Gaussian distribution with zero mean and Pearson correlation coefficient $\rho$" <-- Does this distribution have arbitrary strictly positive variance parameters in both directions? (or all the variances one since the authors refer to "the bivariate Gaussian"?)

p. 5, line 247--248: What does this mean; e.g., what are these "future iterations"? Are the authors saying that the choice of tree and copula family is stable across multiple independent runs of the algorithm? If so, what changes from run to run that this choice might be different? If not, what are the iterations? What is the significance of the lack of change described?

p. 7, line 357: "reduce the variance" <-- The authors might want to use more careful language here to clarify what variance is being reduced since the word "variance" is being used in a few different ways around this point in the text.

p. 8, Table 1 seems to be missing running time. It's in the text, but it's a little hard to parse the comparison in paragraph form. Is there any reason not to put it in the table?

p. 8, line 389: "consistent estimates" <-- This seems to be a poor use of the word "consistent" in the statistical sense (there's no limit as the number of data points becomes infinite). Perhaps "accurate" is the desired word.

p. 8, line 412: This paragraph is a little hard to read. It sounds like CVI took hours, but then the authors say it's better in runtime. Perhaps this could be clarified.

------------------------------ After Author Feedback ------------------------------

The author feedback seemed on point; I look forward to reading more detail about the vine methodology and a broader range of experiments in the final version.
Summary: The authors propose a more accurate approximating set of distributions for variational inference than the mean-field assumption allows by introducing copulas. While their experiments are suggestive of a promising method, explaining the inference of vines and copulas and using a wider variety of models (or at least their existing performance metrics applied across both models they try) would improve the paper.

Author Feedback
Author rebuttal: We thank the reviewers for their constructive feedback. We would like to emphasize that the novelty of the method, which addresses how to efficiently learn the dependency between latent variables without explicit knowledge of the model, has been accepted as valid and legitimate by the reviewers. We are confident this is a useful contribution for making generic inference viable in practice. Omitted comments will be fixed in revision if accepted.

1. R1 questions the applicability of copulas for modelling dependence in a black box framework.

Vine copulas have been successful in modelling dependence for unknown high-dimensional distributions, see, e.g., Schirmacher and Schirmacher (2008), Chollete et al. (2009), Heinen and Valdesogo (2009), de Melo Mendes et al. (2010), Czado et al. (2012), and Nikoloulopoulos et al. (2012).

2. R1 and R2 ask whether the procedure in selecting the vine structure and copula families is sufficient.

The procedure we use for selecting them closely follows the literature (Dissman et al., 2012). Because of the computational infeasibility of searching through all vine structures, we use a greedy approach that sequentially selects each tree based on Kendall's tau. The intuition is that the largest dependencies should appear at the highest-level trees because changes in the copula parameters propagate from top to down during inference; less significant dependencies should appear at the lowest level where they are not as important to accurately estimate. Copula researchers regard this strategy as the most successful approach in practice, and it is used in all current vine software: UNICORN by Ababei et al. (2007), CDVine by Brechman and Schepsmeier (2013), and VineCopula by Schepsmeier et al. (2015). We will add more details in the paper.

3. R1 and R3 ask about the scalability of the method to higher-dimensional problems not present in the experiments.

The algorithm scales linearly with the number of pairwise dependencies modelled by the copula. In the most general case where one incorporates all dependencies, the algorithm is quadratic in the number of latent variables, which is unavoidable. In the examples we consider, the algorithm models dependencies over all local variables, making the dimension of the problems as large as the number of data points itself. The experiments demonstrate its feasibility in such a scenario, as in fact not all copula parameters are updated given a subsample of the data during the stochastic approximations. We will clarify this in the paper.

Regarding models with higher-dimensional global variables, we have studied the mixture of Gaussians where the multivariate normals have dimension 1000. We note that computation time is not affected because the main bottleneck is modelling dependence over the local variables. We will add this experiment to the supplement.

4. R2 asks about the performance metrics in the experiments.

We have looked at predictive accuracy for both examples, c.f., prediction with the mixture of Gaussians for the MNIST digits. Other mixture model parameters in the experiment were omitted from the plot only because of similar results and space constraints. As R1 suggests, we will move details for calculating the gradient into the appendix and, as R2 suggests, use the space for benchmarking other statistics of the posterior, including tails. This will reinforce and better clarify the superiority of our method to LRVB.

5. R1, R2, and R5 ask for comparisons to other algorithms.

R1: Higher order mean-field methods such as TAP are closely related to LRVB which we compare to. They can be seen as a Taylor series correction based on the difference between the posterior and its mean-field approximation (Kappen and Wiegerinck, 2001). As KW (2001) mention, higher order methods assume the mean-field approximation is reliable for the Taylor approximation to make sense, which is not true for all probability distributions and thus is not robust in a black box framework. We also show this sensitivity in our experiments for LRVB; see L393-400. We will comment more on this in the paper.

R1: Other black box methods, e.g., Ranganath et al. (2014), require specification of the variational family. We compare to this in our experiments where the variational family is the mean-field.

R1: HMC is slow in our experiments because it evaluates a gradient over all data points per iteration.

R2: Comparisons to structured approximations will be dependent on the structure imposed by the choice of factorization for each model. Moreover the proposed procedure using copulas does not aim to compete against explicit factorizations, which are not black box; rather it augments them by modelling any dependencies which do not already exist in the factorization.

R5: A full rank Gaussian approximation cannot be compared to as it does not support discrete variables such as the membership assignments in both experiments.